

# The GHGSat-D imaging spectrometer

Dylan Jervis[1], Jason McKeever[1], Berke O. A. Durak[1], James J. Sloan[1], David Gains[1], Daniel J. Varon[1,2], Antoine Ramier[1], Mathias Strupler[1], and Ewan Tarrant[1]

[1]GHGSat, Inc., Montréal, QC H2W 1Y5, Canada.

[2]School of Engineering and Applied Sciences, Harvard University, Cambridge, MA 02138, USA

*Correspondence to*: Dylan Jervis (dylan.jervis@ghgsat.com) and Jason McKeever (jason.mckeever@ghgsat.com)

**Abstract.** The demonstration satellite GHGSat-D or "Claire", launched on June 21, 2016, is the first in a planned constellation of small satellites designed and operated by GHGSat, Inc. to measure greenhouse gas emissions at the facility scale from space. Its instrument measures methane concentrations by collecting and spectrally decomposing solar backscattered radiation in the shortwave infrared using a compact fixed-cavity Fabry-Perot imaging spectrometer. The effective spatial resolution of 50×50 m² over targeted 12×12 km² scenes is unprecedented for a space-based gas sensing spectrometer. Here we report on the instrument design, forward model and retrieval procedure, and present several examples of retrieved methane emissions observed over industrial facilities. We discuss the sources of error limiting the performance of GHGSat-D and identify improvements for our follow-on satellites. Claire's mission has proven that small satellites can be used to identify and quantify methane emissions from industrial facilities, enabling operators to take prompt corrective action.

## 1    Introduction

GHGSat is a Canadian company incorporated in 2011 with the goal to provide a precise, scalable and economical method of measuring greenhouse gas (GHG) emissions from industrial facilities worldwide. The first GHGSat satellite, GHGSat-D or "Claire", is a demonstration small satellite that measures surface-level methane emission plumes with high spatial resolution for facility-scale attribution. The instrument on GHGSat-D is a wide-angle fixed-cavity Fabry-Perot (F-P) imaging spectrometer able to resolve methane ($CH_4$) absorption lines in the shortwave infrared (SWIR) (Sloan et al., 2016), where water ($H_2O$) and carbon dioxide ($CO_2$) absorption lines are also present. The distinguishing features of GHGSat-D compared with other GHG remote sensing missions are its (1) combination of high spatial resolution (~50 m) and fine spectral resolution (~0.1 nm), and (2) compact package (Table 1). To this day GHGSat-D remains the only gas sensing satellite with such high spectral and spatial resolution. This combination enables unique capabilities for imaging and quantifying emission plumes with unambiguous attribution at the facility scale.  Delivering this performance in a compact package enables low-cost launch and operation of the satellite so that a constellation can be used for high-density coverage. To date, GHGSat-D has performed over 5,000 observations of commercial facilities in oil/gas, power generation, coal mining, waste management, and agriculture sectors around the world.



## 1.1 Methane Monitoring for Industry

Methane is a potent greenhouse gas whose concentration has increased from 720 to 1800 ppb since pre-industrial times (Hartmann et al., 2013) and is responsible for a radiative forcing of 0.97 W m$^{-2}$, second only to $CO_2$ (Myhre et al., 2013). Because methane has a relatively short atmospheric lifetime of ~10 years compared to $CO_2$, actions taken that reduce methane emissions will have a significant effect on the near-term warming rate. Anthropogenic methane emissions originate from a very large number of point sources including oil/gas facilities, coal mines, landfills, wastewater treatment plants, and confined livestock operations. Studies have shown that a relatively small fraction of sources are responsible for the majority of methane point source emissions, with 60-90% of overall emissions coming from emitters with flux $Q_0 \geq 100$ kg hr$^{-1}$ (Brandt et al., 2016; Duren et al., 2019). A fleet of satellites with detection threshold at or below $Q_0$ could lead to efficient emission abatement – giving operators the opportunity to take corrective action, often at no net cost (WEO, 2018).

## 1.2 Space-based GHG Monitoring

Facility scale greenhouse gas emissions can be monitored in a variety of ways including: stationary ground-based systems (Chen et al., 2016; Robinson et al., 2011); mobile ground-based measurements (Yacovitch et al., 2015); and aircraft observations (Conley et al., 2016; Duren et al., 2019; Sherwin et al., 2020). Satellites are an attractive complementary observation platform since they have global coverage, employ the same measurement method for any observation site in the world, and can repeatedly revisit any facility in the world. In recent years, building on the pioneering work of the global mapping missions SCIAMACHY, GOSAT, OCO2 and S5P/TROPOMI (Burrows et al., 1995; Hamazaki et al., 2005; Veefkind et al., 2012), satellites have emerged as a candidate technology to measure individual anthropogenic emission sources (Nassar et al., 2017; Pandey et al., 2019). For example, the TROPOMI instrument on board the Sentinel-5P satellite provides daily global coverage with a spatial resolution of several kilometres and can be used to "tip and cue" GHGSat satellites to locate facility scale emission sources. TROPOMI has also been used to detect and quantify very strong emitters (>~10 tonnes hr$^{-1}$), including an industrial "blow-out" event (Pandey et al., 2019). However, until the launch of GHGSat-D spatial resolution had been limited to the kilometre scale and above. Current and future hyperspectral imagers PRISMA (Loizzo et al., 2018), EnMAP (Guanter et al., 2015) and EMIT (Green et al., 2018) have spatial resolution similar to GHGSat (30-60 m) but much coarser spectral resolution (7-10 nm), which limits their detection sensitivity. Recently, the discovery of a number of large methane leaks in an oil/gas production area was reported in (Varon et al., 2019). GHGSat-D's high spatial resolution enabled the attribution of the leaks to specific pieces of equipment within the industrial site, the locations of which were promptly communicated to the site operator. GHGSat-D has also combined multiple single-pass measurements to quantify time-averaged methane emission rates from coal mine vents (Varon et al., 2020).

Whereas satellite instruments can detect the presence of emissions by measuring the column density enhancement relative to background, emission quantification requires a method to model the local transport using meteorological information



(Jongaramrungruang et al., 2019; Nassar et al., 2017; Pandey et al., 2019; Varon et al., 2018). Locally measured meteorological information is usually not available, so wind speed and direction data must be inferred from the plume observations and/or drawn from meteorological databases like the NASA Goddard Earth Observation System Fast Processing (GEOS-FP) reanalysis product (Molod et al., 2012), or similar sources.

## 1.3   **Advantage of High Spatial Resolution**

The primary motivation for a high spatial resolution methane measurement is to identify the industrial facility, or even the piece of infrastructure within the facility, responsible for the detected emissions, to provide actionable information to the operator. High spatial resolution measurements offer additional advantages: (1) the ability to identify most types of clouds within a scene mitigates the need for complex cloud screening algorithms; (2) the ability to image the shape of the emission

plume can help infer wind direction and speed (Jongaramrungruang et al., 2019); and (3) the measured column density enhancement $\Delta\Omega$ in a square ground cell with side length $L$ generally increases for decreasing $L$.

This last point deserves elaboration. The impact of pixel size on the ability to monitor $CO_2$ emissions was studied in (Hill and Nassar, 2019). A useful heuristic introduced by (Jacob et al., 2016) asserts that $\Delta\Omega \propto 1/L$, However, this scaling does not hold

for all plume geometries. Three scenarios are illustrated in Fig. 1., where (b) is the scenario as considered by (Jacob et al., 2016). Here, for constant emission rate and wind speed, the plume enhancement is contained within the pixel boundary along one axis as $L$ is reduced, but "escapes" the pixel boundary along the other axis. Therefore, the amount of excess gas in a pixel of size $L$ scales linearly with $L$, giving the $\Delta\Omega \propto 1/L$ scaling relationship. In Fig. 1(a), on the other hand, the emitted gas remains entirely within the boundary of the smaller pixel $L_1$ leading to an excess mass invariant with $L$, and $\Delta\Omega \propto 1/L^2$. The

scenario in Fig. 1(a) could occur for very low wind speed and/or a transient event where emissions begin just before the satellite observation. Local wind eddies could also produce point-like enhancements. While geometry (a) is likely less common than (b), it is notable since it gives a scaling more favourable than (b) to smaller pixels. Finally, scenario (c) considers a quasi-uniform density field (plume extent greater than $L_2$) – in this case $\Delta\Omega$ is approximately invariant with $L$ and hence there is no advantage to smaller pixel sizes. Geometry (c) might describe widely dispersed plume enhancements far downwind of the

source location or the enhancement from an area emission source. This simplified treatment is based on averaging the excess density within a cell, which neglects the nonlinear character of Beer's law when averaging radiance spectra. When this effect is included, it suppresses $\Delta\Omega$ relative to the simple scaling relationships above when the peak enhancement within a cell significantly exceeds the mean. This leads to an additional, if modest, advantage to smaller pixel sizes.

Space-based methane sensors with kilometre-scale spatial resolutions like TROPOMI have focused on achieving high-precision column measurements ($< 0.95\%$ single measurement) coupled with high absolute accuracy ($< 0.4\%$ single measurement) (Hu et al., 2018). In contrast, a methane sensor with high spatial resolution designed to detect local enhancements above background can have relaxed absolute accuracy requirements. For example, a 5% bias in the absolute



methane column density – on the upper end of what is estimated from neglecting scattering due to aerosols, for instance (Aben et al., 2007; Butz et al., 2009; Houweling et al., 2005) – would lead to a 5% bias in the local methane enhancement and corresponding emission estimate. This error level is much smaller than the total error in a typical emission estimate (Varon et al., 2019), which is usually dominated by wind uncertainty. Furthermore, any additive scene-wide bias in the absolute methane
column density has negligible impact on the ability of the imaging spectrometer to distinguish the local methane enhancement from background and to quantify its magnitude. This holds true even if the additive bias is not consistent between observations – the key requirement is that the retrieval errors not have sharp gradients within the scene.

## 2    Instrument Overview

The GHGSat-D satellite (Table 1) uses the NEMO platform from the University of Toronto Institute for Aerospace Studies
(UTIAS) Space Flight Laboratory. The NEMO platform can host a payload with a mass of up to 6 kg and provide up to 45 W of power. GHGSat-D was launched on Indian Polar Satellite Launch Vehicle C-34 on June 21, 2016 at 23.56 EDT. It was fully commissioned in July 2016 with a 3-5 year expected lifetime. GHGSat-D remains operational at time of writing.

The patented GHGSat-D instrument operates in the shortwave infrared (SWIR) between 1630 – 1675 nm where methane, $CO_2$, and water vapour absorption lines are present. Critical instrument parameters are listed in Table 2. The spectral resolution is
determined by the F-P gap spacing, its uniformity and coating reflectivities. The F-P coating reflectivity is a critical parameter that leads to a trade-off between the per-pixel signal and spectral resolution.

### 2.1    Optical System

The GHGSat-D optical system (Fig. 2) is composed of three lens assemblies with focal lengths $f_{t1}$, $f_{t2}$, and $f_{im}$: the first two lenses, in confocal arrangement, constitute the telescope and the last lens, the imaging assembly, forms a two-dimensional
image of the ground on the detector. The F-P is placed in the Fourier plane of the optical system, between $f_{t2}$ and $f_{im}$. An order sorting filter (OSF), placed between the lenses that make up the $f_{t2}$ assembly, defines the spectral bandpass region. The choice of focal lengths must balance several considerations simultaneously. First, the angular magnification ratio of the telescope $|f_{t1}/f_{t2}|$ must be consistent with the mechanical constraints on input aperture and F-P size, and the spectroscopic constraints that dictate the desired range of incident ray angles on the F-P. Second, the effective focal length $f = f_{im}|f_{t1}/f_{t2}|$
is constrained by the choice of spatial resolution. The ground sampling distance (GSD) of GHGSat-D was chosen to be $L \approx$ 25 m: small enough to resolve facility features, yet large enough to image a ~5 km-sized facility and the full extent of an emission plume. This constrains the effective focal length $f$ of the optical system given the camera pixel size $a$ and orbiting altitude $h$: $f = ah/L \approx 500$ mm. Optical aberrations in the imaging system limit the spatial resolution of GHGSat-D to approximately 50 m.



For a given image, a polychromatic light ray originating from a specific ground location enters the optical system through the input aperture at some angle pair $(\psi, \phi)$, where $\psi$ is the small elevation angle and $\phi$ is the azimuthal angle. In the paraxial approximation, the light ray emerges from the telescope with angle pair $\left(\left|\frac{f_{t1}}{f_{t2}}\right|\psi, \phi\right)$ and is incident on the F-P. The imaging assembly then focuses the light ray to detector pixel

$$(i,j) = \left(i_0 + b\left(\left|\frac{f_{t1}}{f_{t2}}\right|\psi\right)\cos(\phi), j_0 + b\left(\left|\frac{f_{t1}}{f_{t2}}\right|\psi\right)\sin(\phi)\right), \tag{1}$$

where $b$ is a proportionality constant relating angle to pixel radius. The optical axis intercepts the 640 x 512 temperature controlled InGaAs detector array at pixel $(i_0, j_0)$.

## 2.2 Fabry-Perot

The GHGSat-D F-P element consists of two optical flats mounted within a mechanical enclosure. The two optical surfaces are positioned such that the inner surfaces, with reflectivity $R$ and spaced a distance $d$ apart, form an optical cavity. Light with wavelength $\lambda$ and incident angle $\theta = \left|\frac{f_{t1}}{f_{t2}}\right|\psi$ with respect to the F-P surface normal is transmitted according to:

$$T_{FP}(\theta, \lambda) = \frac{1}{1 + \left(\frac{2\,\mathcal{F}}{\pi}\right)^2 \sin^2\left(\frac{2\pi n d \cos(\theta)}{\lambda}\right)} \tag{2}$$

where $n$ is the index of refraction of the medium within the optical cavity and $\mathcal{F} = \pi\sqrt{R}/(1-R)$ is the reflectivity finesse. For each value of $\theta$, the transmission spectrum of the F-P is a series of peaks that are spaced in wavelength by the free spectral range $FSR = \lambda^2/(2d)$ with a spectral width characterized by the full-width half-maximum $FWHM = FSR/\mathcal{F}$. Because the F-P accepts a continuum of $\theta$ values it samples a continuum of wavelengths within the passband.

In applications using F-P based spectrometers, the F-P gap spacing $d$ is often dynamically scanned during the measurement (Reay et al., 1974). In contrast, GHGSat-D uses a fixed gap spacing and exploits the angular dependence of the $m^{th}$ F-P transmission mode's spectral position $\lambda_m = 2\pi n d \cos(\theta)/m$ to measure the spectrum of the incident light. This approach simplifies the mechanical design and makes it much easier to meet stringent stability requirements. The F-P is temperature-controlled to keep thermal mechanical drift to a minimum.

## 3 Measurement Concept

The GHGSat-D spectrometer is based on a wide-angle Fabry-Perot (WAF-P) imaging concept (Sloan et al., 2016). A programmable number of closely overlapping two-dimensional images are taken (typically 200 in nominal operations) in which the atmospheric absorption spectrum is "imprinted" on the images in the form of spectral rings due to the angle-dependent Fabry-Perot transmission spectrum. During the observation sequence, the ground target traverses the field-of-view, sampling the full extent of the spectral information contained in the images. The instrument operates in "target" mode in which





the satellite attitude is adjusted to the keep the facility of interest in the field of view for much longer than it would if operated in nadir (downward pointing) mode, thereby increasing the available integration time, and hence signal-to-noise ratio (SNR).

Fig. 3 illustrates how the F-P samples the backscattered solar radiance spectra to generate spectral rings in the image. Multiple F-P transmission modes are allowed through the OSF bandpass. Because the F-P transmission function depends only on $\theta$, it

is circularly symmetric and so can be expressed as a function of radius $r = \sqrt{(i - i_0)^2 + (j - j_0)^2}$. The $r = 0$ F-P transmission spectrum is shown in Fig. 3(a) alongside the OSF transmission function and the normalized backscattered top-of-atmosphere spectral radiance (TOASR). For larger radii – and thus larger $\theta$ – the F-P spectrum shifts to lower wavelength with a $\cos(\theta)$ dependence, allowing us to sample different regions of the TOASR. Fig. 3(b) shows the location (in wavelength space) of the F-P transmission peaks as a function of radius overlaid on the normalized TOASR. The instrument signal is shown in Fig.

3(c). At each radius, the signal on the detector array is the wavelength integral of the TOASR multiplied by the F-P and OSF transmission spectra, i.e. the result of integrating Fig. 3(b) along the vertical axis. A mathematical description of the forward model is given in Sect. 4.1.

In order to measure the spectrum of solar radiation backscattered from a specific ground cell, the location of the ground cell in each image must be known. This is done with an image co-registration algorithm. We then construct a spectrum for each

ground cell along the image frame axis by recording the measured signal as a function of the ground cell's radial position with respect to the spectral ring center. Fig. 4(a)-(d) show an observation where the location of an example ground cell has been tracked in each frame. The constructed spectrum is shown in Fig. 4(e). The colour of the data point represents the image frame from which the data was obtained. We construct approximately 200,000 of these spectra in order to retrieve the methane column density for each ground cell within the retrieval domain.

**4   Retrieval Method**

The goal of the retrieval algorithm is to estimate the instrument and atmospheric state vector $\mathbf{x}$ from a measurement vector $\boldsymbol{y}$. This is done by constructing a combined forward model $\boldsymbol{F}(\mathbf{x})$ of the instrument and atmosphere and making the association:

$$\boldsymbol{y} = \boldsymbol{F}(\mathbf{x}) + \boldsymbol{\epsilon}_y + \boldsymbol{\epsilon}_F \qquad (3)$$

where $\boldsymbol{\epsilon}_y$ represents the measurement error and $\boldsymbol{\epsilon}_F$ represents error in the forward model. A retrieval of $\mathbf{x}$ requires that we

have accurate knowledge of both the forward model and the errors in the measurement system. Because $\boldsymbol{F}(\mathbf{x})$ is a nonlinear function of $\mathbf{x}$, we must solve for the state vector iteratively. This requires knowledge of the Jacobian $\boldsymbol{K}(\mathbf{x}) = \frac{\partial \boldsymbol{F}(\mathbf{x})}{\partial \mathbf{x}}$ to weight the state vector step $\boldsymbol{\Delta}\mathbf{x}^i$ taken during the $i^{th}$ iteration. In this section, we describe the instrument and atmospheric forward model and outline the inversion procedure used to estimate $\mathbf{x}$.



### 4.1 Forward Model

The forward model represents our best knowledge of the instrument and atmosphere, with approximations used to evaluate the model more efficiently when performing retrievals. The camera signal $F_{i,j}$ at detector pixel $(i, j)$ in photocurrent units [e⁻ s⁻¹] is given by:

$$F_{i,j}(\mathbf{x}) = \int L(\mathbf{x}, \lambda) \cdot C(\lambda) \cdot QE(\lambda) \cdot T_{OSF}(\lambda) \cdot T_{FP}(\theta, \lambda) d\lambda \qquad (4)$$

where $L(\mathbf{x}, \lambda)$ is the spectral radiance as a function of the state parameter $\mathbf{x}$ and wavelength $\lambda$, $C(\lambda)$ is the radiometric conversion factor that converts spectral radiance to the number of photons on a pixel per unit time, $QE(\lambda)$ is the quantum efficiency with which the camera converts a photon to electric charge, $T_{OSF}(\lambda)$ is the transmission of the order-sorting filter that defines the spectral bandpass region, and $T_{FP}(\theta, \lambda)$ is the F-P transmission function defined in Eq. (2). The camera signal in Eq. (4) is plotted as a function of radius in Fig. 3(c).

### 4.2 Atmospheric Model

The spectral radiance $L(\mathbf{x}, \lambda)$ is calculated from the spectral irradiance $I(\lambda)$ assuming Lambertian surface reflectance:

$$L(\mathbf{x}, \lambda) = \frac{a(\lambda) \cdot \cos(\theta_{sza})}{\pi R_{E-S}^2} I(\mathbf{x}', \lambda) \qquad (5)$$

where $a(\lambda)$ is the spectrally-dependent surface albedo, $\theta_{sza}$ is the solar zenith angle, $R_{E-S}$ is the relative Earth-Sun distance, $\mathbf{x}'$ is the state parameter vector without the albedo, and the spectral irradiance is the solution to a simplified radiative transfer equation where thermal emission, aerosol and molecular scattering have been neglected (Chandrasekhar, 2013):

$$\mu \frac{\partial I(\mathbf{x}', \lambda)}{\partial z} = -\alpha_{abs} I(\mathbf{x}', \lambda). \qquad (6)$$

This equation is integrated along the downwelling and upwelling light path. Here $\mu = \cos(\theta)$, $\theta$ is the angle that the light travels through the atmosphere with respect to the Earth's surface normal, $z$ is the altitude, $\alpha_{abs}$ is the pressure, temperature, wavelength, and species dependent absorption coefficient calculated using the HITRAN API (Kochanov et al., 2016), and the solar irradiance is introduced through a boundary condition and generated from the AER solar irradiance model (Clough et al., 2005). We integrate the radiative transfer equation discretely assuming 100 atmospheric layers that are evenly spaced in pressure.

We justify excluding thermal emission and molecular scattering from the atmospheric model because both are small effects for the wavelengths within our spectral bandpass region. Previous studies of simulated carbon dioxide retrievals using only the 1563 - 1585 nm band have found that neglecting aerosol and molecular scattering can lead to a few percent error, depending on the surface albedo and aerosol optical depth (Aben et al., 2007). This error can be either positive or negative, depending on whether the presence of aerosols – in combination with the surface albedo - leads to an increase or decrease in the average optical pathlength. An analysis of SCIAMACHY retrievals in this same spectral region found that errors decrease for aerosol


vertical distributions that are narrow and closer to the Earth's surface (Houweling et al., 2005). As mentioned in Sect. 1.3, GHGSat retrievals are primarily intended to measure local plume enhancements. Therefore, we are especially concerned with any unmodeled effects with spatial structure on the length scales of emission plumes. This could potentially include aerosol scattering, such as aerosols that might conceivably be co-emitted with methane plumes. However, since the presence of these

aerosols plumes would be much closer to the surface and narrower in vertical profile than the aerosol profiles retrieved in (Aben et al., 2007; Houweling et al., 2005), we expect that errors arising from neglecting scattering should be small compared with other sources of measurement error.

### 4.3    Inversion Procedure

For any ground cell $(p, q)$ in a reference frame we can compare the observation data vector $\mathbf{y}^{(pq)} = \{y_{i_1,j_1}^{(pq)}, y_{i_2,j_2}^{(pq)}, \dots, y_{i_k,j_k}^{(pq)}\}$ to

the forward model vector $\mathbf{F}(\mathbf{x}^{(pq)}) = \{F_{i_1,j_1}(\mathbf{x}^{(pq)}), F_{i_2,j_2}(\mathbf{x}^{(pq)}), \dots, F_{i_k,j_k}(\mathbf{x}^{(pq)})\}$ and infer the state vector $\mathbf{x}^{(pq)}$ using a variant of standard inverse methods described below. The subscripts refer to the pixel indices for this ground cell within the respective frames of the image sequence from 1 to $k$. The reference frame coordinate system can then be georeferenced with the appropriate rotational and scale transformation.

We use optimal estimation (Rodgers, 2000) to infer the posterior distribution of the state vector given the observation data, an error model, and a prior distribution for the state vector. Assuming a Gaussian form for the measurement and prior probability density functions, maximizing the joint probability density function amounts to minimizing the cost function:

$$\chi^2(\mathbf{x}) = \left(\mathbf{y} - \mathbf{F}(\mathbf{x})\right)^{\mathrm{T}} \mathbf{S_o}^{-1}\left(\mathbf{y} - \mathbf{F}(\mathbf{x})\right) + (\mathbf{x} - \mathbf{x}_a)^{\mathrm{T}} \mathbf{S_a}^{-1}(\mathbf{x} - \mathbf{x}_a) \tag{7}$$

where $\mathbf{S_o}$ is the observation error covariance matrix, $\mathbf{S_a}$ is the prior covariance matrix, and $\mathbf{x}_a$ is the prior state vector. The

Gauss-Newton procedure for minimizing the cost function requires that we update the state vector at each iteration by a step:

$$\Delta \mathbf{x}^{i+1} = (\mathbf{K}^{\mathrm{T}}(\mathbf{x}^i)\mathbf{S_o}^{-1}\mathbf{K}(\mathbf{x}^i) + \mathbf{S_a}^{-1})^{-1} \left(\mathbf{K}^{\mathrm{T}}(\mathbf{x}^i)\mathbf{S_o}^{-1}\left(\mathbf{y} - \mathbf{F}(\mathbf{x}^i)\right) + \mathbf{S_a}^{-1}(\mathbf{x} - \mathbf{x}_a)\right) \tag{8}$$

where $\mathbf{K}(\mathbf{x}^i)$ is the Jacobian of the forward model evaluated at $\mathbf{x}^i$.

At each iteration of the Gauss-Newton procedure, the forward model and Jacobian must be evaluated. This is computationally

expensive for a single cell and evaluating it for the ~200,000 ground cells in our field-of-view is impractical. Instead, we use a two-step procedure: (1) a scene-wide average retrieval using the full forward model to estimate the scene-wide average state vector $\hat{\mathbf{x}}$, and (2) a per-cell retrieval done using a linearized forward model (LFM) evaluated at the linearization point $\hat{\mathbf{x}}$. The separate retrieval steps are described in the following sections.





## 4.4 Scene-wide Retrieval

The goal of the scene-wide retrieval is two-fold: to retrieve scene-wide averaged surface and atmospheric parameters such as albedo and molecular column density, and to retrieve the F-P gap spacing. Even though the F-P is thermally stabilized, residual drift in the F-P gap spacing can occur between observations. Because the F-P gap spacing directly affects the signal on each detector pixel, we retrieve $d$ for each observation. The scene-wide retrieval uses the full instrument model $F_{i,j}(\mathbf{x})$ from Eq. (4) in the optimal estimation procedure. The data vector $\mathbf{y}$ in the scene-wide retrieval is the radial average of the average of all image frames. The prior $\mathbf{x}_a$ uses same-scene information from Landsat-8 for the albedo parameter (Roy et al., 2014) and closest-in-time methane, $CO_2$, and water vapour values from AIRS (Chahine et al., 2006). The result of the scene-wide retrieval is the state parameter estimate $\hat{\mathbf{x}}$.

## 4.5 Spatially Resolved Column Retrieval

A linearized forward model (LFM) is constructed at the linearization point $\hat{\mathbf{x}}$:

$$\mathbf{F}^{LFM}(\mathbf{x}^{(pq)}) = \mathbf{x}^{(pq)}(1)\left(\mathbf{K}_1(\hat{\mathbf{x}}) + \frac{1}{\hat{\mathbf{x}}(1)}\sum_{l=2}^{n}\mathbf{K}_l(\hat{\mathbf{x}})\left(\mathbf{x}^{(pq)}(l) - \hat{\mathbf{x}}(l)\right)\right) + \sum_{l=n}^{n+m}\mathbf{K}_l(\hat{\mathbf{x}})\left(\mathbf{x}^{(pq)}(l) - \hat{\mathbf{x}}(l)\right) \qquad (9)$$

where the first element of the state vector is taken to be the surface reflectance. There are two terms in the LFM: one with $n$ terms that includes state parameters whose Jacobians scale with the surface reflectance (for example the molecular column densities), and another with $m$ terms for state parameters whose Jacobians are not scaled by the reflectance. The primary advantage of using the LFM is that we only compute the forward model and Jacobians once at the beginning of the per-cell retrieval. A disadvantage is that for the parameters that are nonlinear in the forward model, a retrieval using the LFM will introduce systematic biases for deviations far from the linearization point. For the particular case of molecular column densities, this leads to an underestimation that is corrected in post-processing using a non-linear correction function determined from a comparison of $\mathbf{F}^{LFM}(\mathbf{x})$ with $\mathbf{F}(\mathbf{x})$ at the appropriate linearization point (Varon et al., 2019).

## 4.6 Data Processing

The inversion procedure is performed after data downlinked from the satellite has been processed to the following data levels (Kobler et al., 1995):

- Level 0: information received from the satellite is removed of all communication related artefacts. Telemetry data is parsed and stored separately from the image observation data.
- Level 1A: Telemetry data is processed to provide instrument position, orientation, solar zenith and observation angles, etc..
- Level 1B: The recorded ADC values in the image observation are converted to photocurrent units [e⁻ s⁻¹] after correction for pixel offset and dark current. Corrections are also applied to mitigate optical ghosting and detect and flag dead, hot, or otherwise misbehaving pixels.



Because GHGSat-D does not contain an on-board calibration unit, changes in pixel offset signal and dark current are measured by taking frequent observations over a dark ocean scene. The observation data vector $\mathbf{y}$ that the optimal estimation procedure

is performed on is then the Level 1B data that has been "dark" corrected by the most recent ocean measurements. The optimal estimation procedure generates the spatially resolved state vector elements $\mathbf{x}^{(pq)}$. These are then used to generate the Level 2 product, consisting of georeferenced arrays of methane abundances and their uncertainties.

## 5    Data and Measurement Performance

An example comparison between data and retrieved forward model from a single 24 x 24 $m^2$ ground cell is shown in Fig. 4(e).

The ground cell samples approximately the same radius – and thus the same wavelengths - twice as the ground location traverses the imaging field-of-view. However, it is evident that the signal at the same radius can have different values. This is primarily due to two effects: (1) the bidirectional scattering distribution function of the piece of terrain within a ground cell is sampled at two different observation angles during the measurement sequence, and (2) per-pixel signal changes due to satellite motion during the image integration time. The forward model accounts for this slow signal variation by replacing the constant

albedo with a $2^{nd}$-order polynomial that is a function of the image frame index, similar in concept to what is done in DOAS retrievals (Platt, 1994) where the polynomial is a function of wavelength. The forward model also accounts for the fact that the ground cell overlap with a given camera pixel changes between different frames during the observation sequence. Knowledge of the ground cell/camera pixel overlap is provided by the image co-registration algorithm and allows us to account for data effects that, if not treated, would result in erroneous high-frequency error. Residuals between data and the retrieved

forward model with a standard deviation of 0.5% are representative.

When the methane enhancement component of the state vector $\mathbf{x}$ is retrieved from the ~200,000 ground cells within a measurement domain, a retrieved methane map can be plotted as in Fig. 5, which shows a selection of GHGSat-D measurements taken over various types of industrial sites around the world. In each retrieved methane enhancement field, we

observed a localized methane plume whose point of origin coincides with a facility. For each observation, the plume is also displayed as an overlay on the retrieved SWIR surface reflectance image using thresholding and a spatial correlation criterion that counts a downwind enhancement as real if it is close (within a few pixels) to another enhancement that was previously determined to be real, with the requirement that the furthest upwind enhancement closely overlaps the location of a facility source. The retrieved SWIR reflectance can then be used to geolocate the retrieved methane enhancement field. The units of

the excess methane vertical column density (VCD) are [mol $m^{-2}$]. For comparison, the nominal VCD background value is approximately 0.67 mol $m^{-2}$, corresponding to a column averaged mixing ratio of XCH4 $\approx$ 1.9 ppm. The peak plume enhancements in the examples presented are 20-120% above background. In each observation, the origin of the methane plume



can be clearly attributed to a ground location with an uncertainty of approximately 30 m. This location accuracy is sufficient to provide actionable information to facility operators.

In the example observations shown in Fig. 5, methane plumes were observed over a variety of industrial facilities: a hydro-
electric reservoir, coal mine vents, and natural gas sites. On April 20th, 2017 we observed a methane plume over the dam vanes of the Lom Pangar hydro-electric reservoir (Fig. 5(a)) in eastern Cameroon that was flooded the previous year. Hydro-electric reservoirs are a known source of methane and carbon dioxide emissions, especially those in tropical climates that have been recently flooded (Barros et al., 2011; Rosa et al., 2004). On October 17th, 2018 we observed a methane plume over a natural gas facility in the Permian basin, TX, USA (Fig. 5(b)). This was a suspected liquid unloading event in which liquid in the well
is removed to keep gas flowing to surface facilities, often resulting in a large, but temporary, pulse of methane emissions. On October 18th, 2018 and September 18th, 2018 we observed methane plumes over vents in coal mining operations near Camden, NSW, Australia (Fig. 5(c)) and Farmington, NM, USA (Fig. 5(d)), respectively. Methane can be released from coal and surrounding rock strata during mining operations and large methane emissions have been observed for both of these sites in previous studies (Frankenberg et al., 2016; Ong et al., 2017). In the Farmington, NM observation, we can see that the magnitude
of the plume enhancement is commensurate with those of neighbouring enhancement artefacts in the methane retrieval field. This fact highlights the advantage of incorporating prior information about source locations when determining whether a measured enhancement at or near the detection limit is real. On February 24th, 2019 and March 9th, 2019 we observed large methane plumes over the Korpezhe oil/gas field in western Turkmenistan (Fig. 5(e) and (f)). Large methane emissions from this area have been previously reported in (Varon et al., 2019).

The error in methane retrieved from a single-pass GHGSat-D observation is typically between 8-25% of the background value, depending on factors such as the time of year and complexity of the albedo field. This error is due to various GHGSat-D instrument imperfections, including out-of-field stray light, in-field optical ghosting, and memory lag effects in the camera response. Minor imperfections and uncertainties in our instrument model can lead to significant systematic errors in the
methane retrievals. One of the most obvious errors in the methane retrieval is "streaking" in the direction of the along-track satellite motion. The high-frequency streaking is likely due to camera pixels which are dead, "hot" (always on), blinking, or have mischaracterized offset or gain that have not been flagged by the bad pixel detector. The low-frequency character of the streaking can be explained by unwanted optical signal (from straylight or ghosting) coupling to the spectral dips, especially those that occur near tangents of the spectral rings.


It is useful to compare the observed measurement error levels to the limit set by random noise on the camera. The GHGSat-D per-pixel signal-to-noise ratio is $\text{SNR} = I_0/\sigma = 200$, with $I_0$ the mean per-pixel signal level and $\sigma^2$ the sum of the mean photon shot-noise, camera dark noise and read noise variance. To compare to the methane measurement error, we define the methane SNR, $\text{SNR}_{CH_4}$, using the posterior error $\widehat{S} = (K^T S_o^{-1} K + S_a^{-1})^{-1}$ derived from optimal estimation theory (Rodgers,


2000) which has a diagonal error covariance matrix $\boldsymbol{S_0}$ populated with $\sigma^2$. Our definition of $\text{SNR}_{CH_4}$ is the inverse square root of the molecule state parameter element in $\widehat{\boldsymbol{S}}^{-1}$ with all elements of $\boldsymbol{S_a}$ set to zero except for methane (which is set to infinity – i.e. the estimate for this parameter is determined solely from the data). We then get:

$$\text{SNR}_{CH_4} = \Omega_{CH_4} \cdot \left( \sigma^2 \left( \boldsymbol{K}_{CH_4}^T \cdot \boldsymbol{K}_{CH_4} \right)^{-1} \right)^{-\frac{1}{2}} = \text{SNR} \cdot \Omega_{CH_4} \cdot \left( \boldsymbol{k}_{CH_4}^T \cdot \boldsymbol{k}_{CH_4} \right)^{\frac{1}{2}} \tag{10}$$

where $\Omega_{CH_4}$ is the total methane VCD, $\boldsymbol{K}_{CH_4}$ is the methane Jacobian, and $\boldsymbol{k}_{CH_4} = \boldsymbol{K}_{CH_4}/I_0$ is the Jacobian normalized by signal level. This expression can also be derived from a linear least-squares method (Adler et al., 2010). For a nominal albedo of 0.2, solar zenith angle of 40°, and US-Standard temperature, pressure, and mixing ratio profiles (Coesa, 1976) scaled to present values of the total column densities, $\text{SNR}_{CH_4} = 15.4$. The 8-25% methane measurement error can be compared to the ideal limit $\text{SNR}_{CH_4}^{-1} = 6.5\ \%$. We see that the lowest error GHGSat-D observations approach the ideal limit. These tend to be

observations over bright scenes with quasi-uniform surface reflectance, such as deserts or plains. Conversely, observations with larger error in the methane retrieval tend to occur over areas with highly non-uniform and/or low surface reflectance, such as urban scenes.

Given the range of column error levels of 8-25%, our experience with source rate retrievals (Varon et al., 2018, 2019) suggests

that GHGSat-D is sensitive to point emitters with $Q > 1000 - 3000$ kg hr$^{-1}$. Using a typical value of wind speed (3 m s$^{-1}$) this is consistent with a simple model for point source detection threshold based on excess methane in the source pixel column (Jacob et al., 2016).

## 6  Summary and Future Plans

GHGSat has developed, built, and successfully launched a demonstration satellite, GHGSat-D, continuously operational since

2016. GHGSat-D uses a wide-angle Fabry-Perot imaging spectrometer to make quantitative measurements of the methane column density, with a focus on resolving enhancements above background values within the 12 x 12 km$^2$ measurement field-of-view. Since beginning on-orbit measurements in 2016, GHGSat-D has made over 5,000 observations including several demonstrated discoveries of industrial methane emissions from space (Varon et al., 2019). This mission has proven that a compact spectrometer on a small satellite can be used to detect and quantify methane plumes from individual facilities with

unambiguous attribution. While its detection threshold is estimated to be relatively high at 1000-3000 kg hr$^{-1}$, the experience of designing, manufacturing and operating GHGSat-D has been a highly fruitful process. Detailed investigations of the retrieval outputs and comparison with simulations under various conditions have helped us understand the limiting sources of error and informed the design of our next satellites, which have much better projected performance.

The first satellite in GHGSat's commercial constellation, GHGSat-C1 or "Iris", was successfully launched on Sept 2$^{nd}$, 2020. The second commercial satellite, GHGSat-C2 or "Hugo", is scheduled to launch in December 2020. At the time of this





manuscript's submission, GHGSat-C1 was in its commissioning phase and had detected its first emission plumes from industrial facilities. GHGSat-C1 has an improved design informed by lessons learned from GHGSat-D. Most importantly, GHGSat-C1 has a 100x reduction (approx.) in straylight magnitude, a 5x reduction in ghosting magnitude, increased per-pixel signal levels, an on-board dark and flat-field calibration system, and a re-optimized spectroscopic configuration. GHGSat-C1

has also undergone an intensive test and characterization campaign in which camera and instrument behaviour have been more extensively explored, calibrated, and parameterized than was done for GHGSat-D. The retrieval method has also been advanced, including significant improvements in alignment and spatially resolved column retrievals, tested using aircraft trials ahead of the launch of GHGSat-C1. GHGSat will perform a calibration and validation campaign for GHGSat-C1 that includes, among other activities, several controlled methane release campaigns. We estimate that we will achieve column errors of ~2%

with GHGSat-C1 - including systematic errors - for a nominal observation (subsequent satellites will have similar or better performance). Given these improvements in the column precision and finer spatial resolution (~25 m expected), we anticipate satellites in our constellation to achieve detection thresholds at or below 100 kg hr$^{-1}$ for nominal conditions. As we scale up the constellation, this will allow us to provide increasing amounts of high-fidelity, actionable data to industrial operators worldwide, ultimately leading to significant emissions reductions.

**Author contributions:**

JM, DJ, BOAD, JJS, and DG developed and implemented the measurement and retrieval concept. JM, DJ, MS and AR analyzed the instrument characterization data. ET contributed improvements to the retrieval method. DV developed source rate retrieval methods. DJ and JM wrote the manuscript with comments and revisions from all authors.

**Acknowledgements**:

We would like to thank the GHGSat Operations team for processing the large volume of data generated by GHGSat-D observations. We would like to thank MPB communications and the Space Flight Laboratory for the close collaborations developed through the design, construction, and daily operation of GHGSat-D. We thank Daniel Jacob, Ilse Aben, and Ruud

Hoogeven for helpful comments during the writing of the manuscript.

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



**Table 1: Satellite Parameters for GHGSat-D**

| Parameter | Value |
|---|---|
| Satellite mass | 15 kg |
| Satellite dimensions | $20 \times 30 \times 40$ cm$^3$ |
| Payload dimensions | $12 \times 12 \times 25$ cm$^3$ |
| Launch date | June 21, 2016 |
| Orbit type | Polar, sun-synchronous |
| Local time at descending node | 09:30 |
| Altitude | 514 km |



**Table 2: Instrument parameters**

| Parameter | Value | Comments |
|---|---|---|
| Pixel size [$\mu$m] | 25 | |
| Camera array dimensions | 640 x 512 | |
| Spectral range [nm] | 1630 – 1675 | |
| Spectral resolution [nm] | ~0.1 | FWHM of each F-P transmission mode. Note that multiple F-P modes contribute signal at each pixel, leading in many cases to an effective coarsening of the spectral resolution. |
| Spectral sampling [nm/pixel] | 0.0001 – 0.1 | Spectral sampling is nonlinear across the detector due to F-P transmission mode behaviour. Spectral sampling is finer near the center of the detector and coarser at greater radii. |
| Ground sampling distance (GSD) [m] | 24 | At altitude 514 km (Table 1)) |
| Spatial resolution [$m^2$] | 50 x 50 | Effective resolution is coarser than GSD due to optical aberrations and other effects. |
| Field of view diameter [km] | 12 | Illuminated portion of each frame is circular |
| Methane retrieval domain size [$km^2$] | $\geq 12 \times 12$ | Since the retrievals are derived from image sequences with a programmable degree of overlap, the retrieval domain differs in shape from the imaging FOV. |
| SNR (typical) | 200 | Defined as the per-pixel signal for a 0.2 albedo scene and solar zenith angle of 40° divided by the shot noise, dark noise, and read noise. |
| Methane SNR (typical) | 15.4 | Defined from SNR using optimal estimation theory in Sect. 4. A theoretical performance limit based on random noise (not including systematics). |



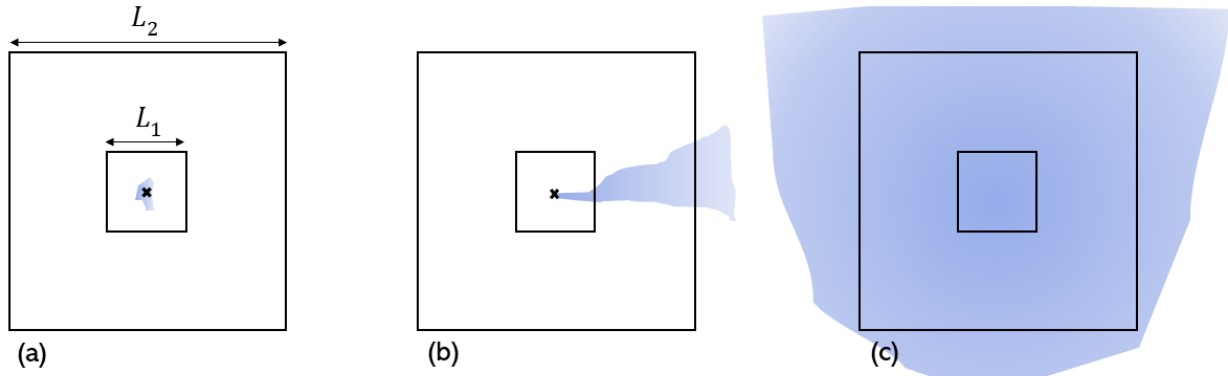

**Figure 1:** Plume geometry scenarios for illustrating the dependence of detected enhancement $\Delta\Omega$ on pixel size $L$.





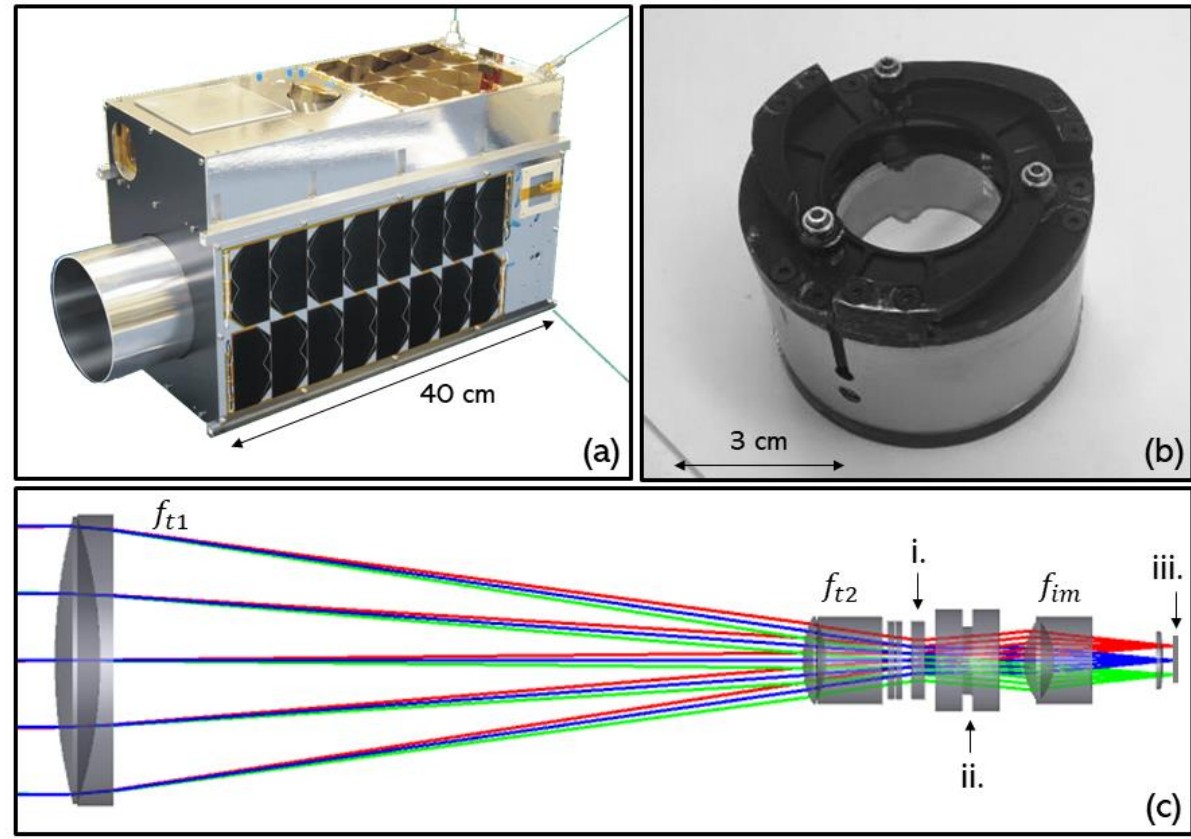

**Figure 2.** (a) The GHGSat-D spacecraft with the imaging spectrometer onboard. (b) The mounted Fabry-Perot interferometer. (c) Schematic of the unfolded optical system with the i.) OSF, ii.) F-P, and iii.) detector identified. The red, blue, and green rays originate from different ground locations.

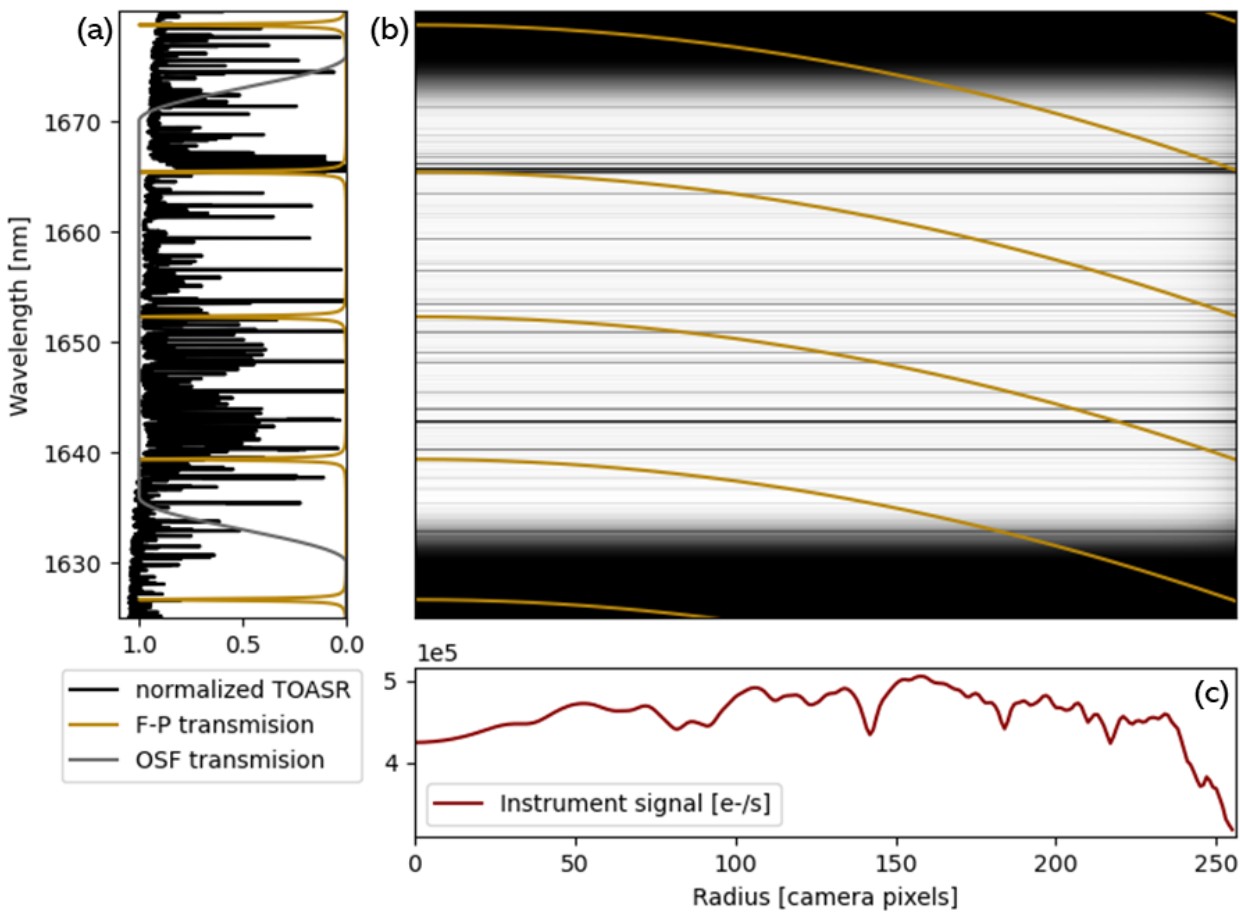

**Figure 3:** The plots (a) and (b) illustrate how the instrument signal in (c) is produced. In (a), the F-P and OSF transmission spectra are shown alongside the normalized top-of-atmosphere spectral radiance (TOASR) for the case of light rays incident on the F-P at normal incidence ( i.e. $(r, \theta) = 0$). In (b), the location of the F-P transmission peaks are shown as a function of radius (gold lines) overlaid on the normalized TOASR (grey-scale background image). The horizontal dark bands at the top and bottom of (b) illustrate wavelengths where the OSF transmission is reduced to zero. By sampling a continuum of incident angles $\theta$, the transmitted F-P transmission peaks measure a continuum if wavelengths within the passband. The instrument signal (c) results from integrating the multiplied signals in (b) along the vertical (wavelength) axis.



**Figure 4:** The images (a), (b), (c)., and (d). show a selection of frames from an observation over the Lom Pangar hydroelectric reservoir in Cameroon taken on April 20[th]., 2017 with an example ground location (denoted by an orange "x") tracked in each frame. The image axes are in pixels, with each pixel representing a 24 x 24 $m^2$ area on the ground. The plot in (e) shows the signal (circles) from the example ground location as a function of the image frame (circle colour) and radius from spectral ring center (horizontal axis). The forward model (black line) is plotted alongside the signal data and residuals between model and data are shown in (f).

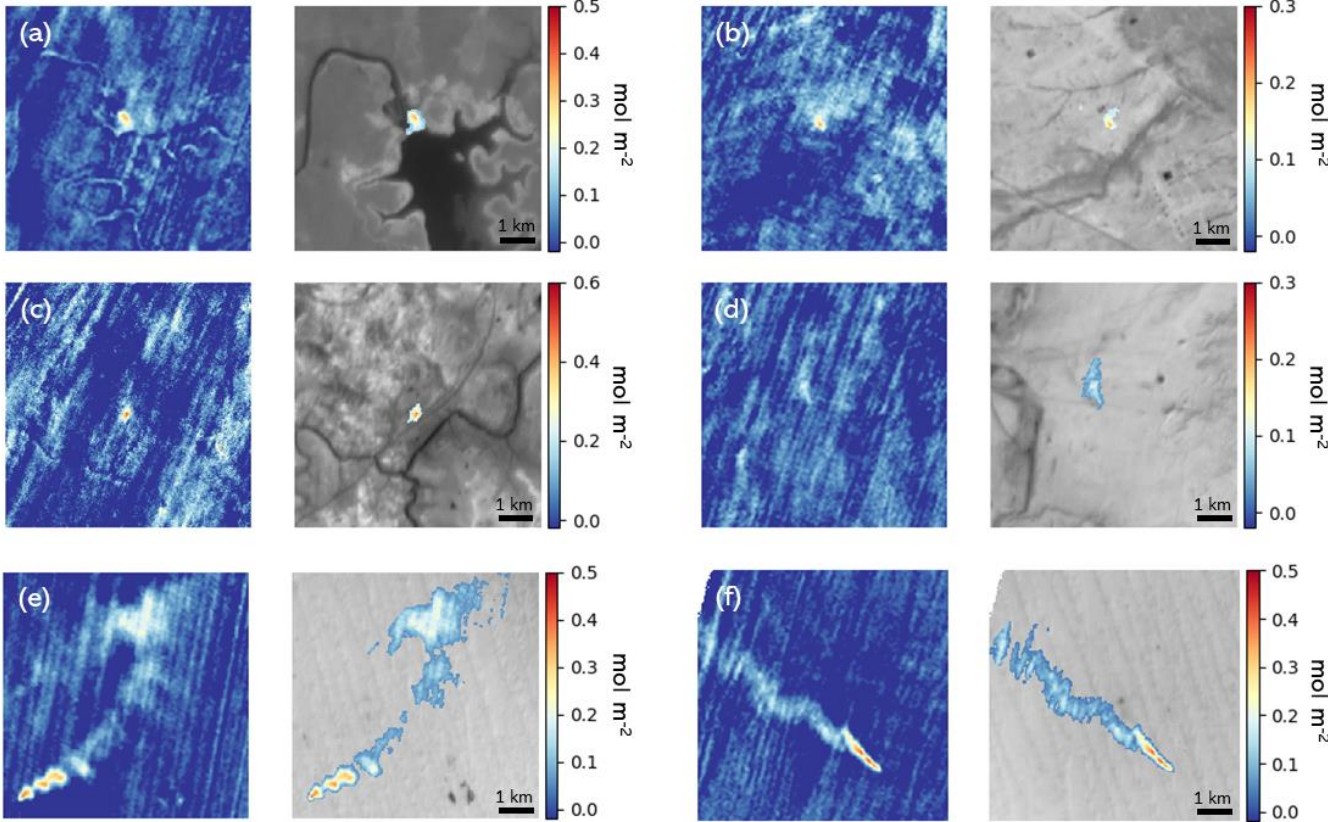

**Figure 5:** Retrieved methane enhancement fields. In the left plot of each plot-pair, the methane enhancement above the local background value is shown in a 7 x 7 km² region of interest centred on the plume. In the right plot of each plot-pair, the extracted plume is overlaid on top the retrieved surface reflectance. The methane enhancement colour scale is in units of mol m⁻². Description, location, and date of observations: (a) hydroelectric reservoir at the Lom Pangar dam, Cameroon, April 20[th], 2017; (b) suspected liquid unloading event in the Permian basin, Texas, USA, October 17[th], 2018; (c) underground coal mine vent near Camden, Australia, October 18[th], 2018; (d) underground coal mine vent near Farmington, New Mexico, USA, September 18[th], 2018; (e) natural gas compressor facility near Korpezhe, Turkmenistan, February 24[th], 2019; (f) same natural gas compressor facility near Korpezhe, Turkmenistan, March 9[th], 2019. In all retrieval fields, the plume enhancement competes against retrieval artefacts that are usually oriented along the direction of the satellite orbital direction.