# Peer review of "The GHGSat-D imaging spectrometer"

_Atmospheric Measurement Techniques, 2020_

## Short Comment (SC1) · 12 Oct 2020

Thanks to the authors for these commendable results and interesting discussion. I have a brief suggestion in the interest of improving the manuscript. Line 22 states of facility-scale GHG monitoring that "until the launch of GHGSat-D spatial resolution had been limited to the kilometre scale and above." However, $CH_4$ plume images at 30m spatial resolution from the Hyperion imaging spectrometer had been published in May 2016 (https://doi.org/10.1002/2016GL069079).
* * *

---

## Referee Comment (RC1) · Anonymous Referee #1 · 19 Oct 2020

General comments: Jervis et al. describe a space-borne imaging spectrometer using a fixed-cavity Fabry Perot interferometer (FP) for wavelength selection at around 1650nm. The FP, together with an order sorting filter is placed inside a camera optics, causing rings of equal FP transmission on the detector array through the FP's incidence angle dependence. Thereby, in a series of recorded images, points within the FOV are observed with several different FP spectral transmissions through the satellite movement. The resulting interferograms are evaluated for methane absorption by inversion of a combined instrument and atmospheric model. The high spatial resolution allows for detecting very strong methane gradients, for instance emissions (e.g. leaks) from industrial facilities. The paper has a clear structure and fits into the scope of AMT. I have two major points:

1) Scattering at aerosol is neglected in the atmospheric model motivated by e.g. the work of Houweling et al., 2005, which treats total column $CO_2$ measurements. I think

that the finding of a decreased aerosol induced error for narrow layers close to Earth's surface cannot directly be transferred to the presented study of localised and strong emission plumes with much higher spatial resolution. The presented methane emission measurements show a different geometry with very high concentrations at low altitude. For low altitude aerosol and particularly for co-emitted aerosol (as mentioned in the manuscript, l.4,p.8), induced light path changes are likely to have a stronger impact on the vertical column density quantification. And thus, the influence of aerosol on the flux determination can become relevant.

2) The authors mention the absorption of CO2 and water vapour in the chosen wave-length window (p4, l13-14). If CO2 and water vapour amount are fixed parameters in the model inversion, their cross interferences need to be quantified in order to exclude any significant influence on the methane measurement. The influence of local gradients (e.g. emission plumes or co-emission) of these gases, as well as aerosol induced light path variations (see above) should be quantified (e.g. by using the introduced model).

Specific Comments

3) Figure. 3 (a) indicates that within the pass band of the order sorting filter there are three FP transmission fringes. This should be mentioned/motivated in the instrument description. Selecting 3 transmission peaks triples the light throughput compared to a measurement with a single peak and therefore enhances the SNR by sqrt(3). On the other hand, a dilution of the absorption signal of strong absorption lines is expected, reducing the SNR by up to a factor of 3. For the FP's free spectral range correlating with the spectral separation of strong periodic absorption structures (e.g. as in Vargas-Rodriguez and Rutt, 2009 or Kuhn et al., 2019) the sensitivity, selectivity and the SNR would be increased by using several FP transmission peaks. The instrument description does not mention if such a correlation is used.

4) How does the measurement error/sensitivity vary across the imaging FOV? In Fig.

4 (a)-(d) the rings of equal FP transmission are faintly visible. A pixel located in the centre of the detector will see a different FP interferogram compared to a pixel close to the detector edge. I.e. the progression of the signal as shown in Fig. 4 (e) is dependent on the location of a pixel on the detector. A slight tilt of the FP in along track direction would increase the radii of the FP rings within the FOV. This could have the advantage of a better coverage of the whole FOV with similar FP interferograms and also it would increase the range of FP tuning ($\sim$cos(alpha)) per pixel. Thereby the spectral information of the measurement could be further enhanced and areas on the detector with low dynamics in spectral FP changes (e.g. the centre area of the detector) were avoided.

5) Fig. 3 (c) shows the 'instrument signal' as a function of the radius. Here it would be illustrative to show the differential instrument signal between a typical methane plume and a plume free region. Thereby the sensitivity of the method in terms of measured optical depth per methane amount would become more clear. Also the influence of typical $CO_2$ and water vapour absorption gradients could be illustrated that way.

References

Vargas-Rodríguez, E. and Rutt, H.: Design of CO, $CO_2$ and $CH_4$ gas sensors based on correlation spectroscopy using a Fabry– Perot interferometer, Sensor. Actuat. B Chem., 137, 410–419, https://doi.org/10.1016/j.snb.2009.01.013, 2009.

Kuhn, J., Platt, U., Bobrowski, N., and Wagner, T.: Towards imaging of atmospheric trace gases using Fabry–Pérot interferometer correlation spectroscopy in the UV and visible spectral range, Atmos. Meas. Tech., 12, 735–747, https://doi.org/10.5194/amt-12-735-2019, 2019.

---

## Referee Comment (RC2) · Anonymous Referee #2 · 24 Nov 2020

This paper outlines the design, retrieval algorithm, and performance of the GHGSat-D instrument, a Fabry-Perot spectrometer to measure methane columns at high spatial resolution (~50 m). The paper is laid out and written well. The instrument seems to be a clever design, though this first instrument appears to suffer from some serious instrument artefacts, as are described in the paper.

I recommend publication after minor revisions as outlined below. My main suggestion is to include more information on the retrieval algorithm and the implicit assumptions of your forward model and prior uncertainties.

**Specific Comments**
P2, L25: "…which limits their detection sensitivity." Can you state any more quantitatively what the detection limit of these coarse spectral resolution instruments might be? Even providing a range would be helpful to the non-expert (like me).

P7, L20: Please state which version of HITRAN you use (2012, 2016, etc) and if you have done any sensitivity tests to see the effect on your retrievals using different versions.

P7, L21: It's becoming "standard" in our community to use the "Geoff Toon" solar transmittance spectrum. Mainly because everything thinks Geoff is incredibly careful and pays more attention to detail than anyone else. In any case, typically the best residuals are yielded when you use this linelist. It may not matter much for your application, but just FYI. (https://mark4sun.jpl.nasa.gov/toon/solar/solar_spectrum.html, I recommend 2016, disk-integrated spectrum). You have to multiply by the solar continuum yourself.

P8, L25: I do not understand why it is so computationally expensive. Your forward model is literally Fsun(lambda) * mu0 * Albedo(lambda) /pi * exp(-m * tau(lambda)), where m is the airmass factor. Nearly all the work is done in calculating tau(lambda):
tau(lambda) = tau_ch4(lambda) + tau_h2o(lambda)
tau_ch4(lambda) = Sum_i=1..Nlayers{ sigma_ch4(i,lambda) * uch4(i) }
tau_h2o(lambda) = Sum_i=1..Nlayers{ sigma_h2o(i,lambda) * uh2o(i) }

The sigma's can be calculated once for the whole scene (unless you're retrieving surface pressure or temperature, which I do not think you are).

P9, L5: Is retrieving *d* analogous to retrieving the wavelength shift and stretch for a standard GOSAT-type retrieval? If so, say so.

P9, Sect 4.5: It is hard to tell what you are doing without more information – could you please include a table that lists your state vector parameters, the source of your prior information or the prior values themselves, and the prior uncertainty you assume? Also, do you assume any spatial pixel-to-pixel correlations in your prior covariance matrix? Do you retrieve a scale factor

for CH4, CH4 in the boundary layer only, or a full profile in some way?  Etc.  {It looks like you retrieve "VCD".  But do you spread this equally in the vertical?  Ie, you might have sensitivity to HOW you retrieve CH4.  What if you put it all in the boundary layer?  Because the gas absorption cross section is different down there, I expect it will affect your results in a small way.}

Also, I'm surprised you can use AIRS.  Plus AIRS will die someday.  And all the decent NWP models assimilate it anyway.  Why not just use GEOS5 or ECMWF model output?

Additionally, where does your prior surface pressure and temperature profile come from?  Do you assume a fixed surface elevation across the field of view, or do you use a high-res DEM to get the surface elevation for each pixel, and a hypsometric adjustment to determine the prior surface pressure of each pixel?  You don't say anything about if you can do this over complex topography or not.

Finally, can you say how many iterations this takes typically, and how long the retrieval takes for one full scene of data (on however many CPUs you typically use).

Same paragraph – do you have any way to test for and remove bad pixels?  You could also attempt to include a stray light parameter or two in your forward model.  You'd probably have the information content to solve for it.

P12, L10: I'm really surprised by this 6.5% theoretical (noise-drive) error.  This is something like 120 ppb of methane.  TROPOMI has errors more on the order of 10 ppb.  And your number assumes PERFECT knowledge of temperature, water vapor, surface albedo, surface pressure, spectroscopy, instrument, etc.  Literally everything but the instrument noise, which isn't bad; you list radiance SNR = 200 at these conditions.  I have a feeling that if I wrote my own simple Beer's law forward model in wavelength space, and put an SNR of 200 per channel, using your spectral resolving power, I'd get a posterior uncertainty of less than 1%.   What am I missing?  In other words, can you say why this noise-driven uncertainty is so large compared to what a grating instrument with a similar SNR and spectral resolution would achieve?

**Technical**
L26: "reported in (Varon et al., 2019)" should be "reported in Varon et al. (2019)".   Use \citet instead of \citep in LaTeX.  Similar for P3, L13-15 and various places throughout the document.

---

## Author Comment (AC1) · 21 Dec 2020

We thank the author for reminding us of this important paper. We have included a mention and citation of it in the revised manuscript.

---

## Author Comment (AC2) · 21 Dec 2020

**Responses to reviewers: "The GHGSat-D imaging spectrometer"**

We thank the reviewers for their comments and suggestions, which we address below. Reviewer comments are in *italics* and our responses are in normal font style.

**Anonymous Referee #1**

1) *Scattering at aerosol is neglected in the atmospheric model motivated by e.g. the work of Houweling et al., 2005, which treats total column CO2 measurements. I think that the finding of a decreased aerosol induced error for narrow layers close to Earth's surface cannot directly be transferred to the presented study of localised and strong emission plumes with much higher spatial resolution. The presented methane emission measurements show a different geometry with very high concentrations at low altitude. For low altitude aerosol and particularly for co-emitted aerosol (as mentioned in the manuscript, l.4,p.8), induced light path changes are likely to have a stronger impact on the vertical column density quantification. And thus, the influence of aerosol on the flux determination can become relevant.*

We agree with the reviewer that the case studied by (Houweling et al., 2005) is similar, but not exactly the same, to the case of a localized aerosol plume co-emitted with methane. Therefore, we have included a citation to a recent study (Huang et al., 2020) showing that neglecting aerosols leads to approximately 5% error for cases of a co-emitted aerosol plume with significant aerosol optical depth, an error level that is small compared to other errors that afflict our measurement (8-25%, as reported in Section 5). We have also included a citation of a AVIRIS methane retrieval paper (Thorpe et al., 2014) which justifies neglecting aerosol scattering in the forward model on similar grounds.

2) *The authors mention the absorption of CO2 and water vapour in the chosen wavelength window (p4, l13-14). If CO2 and water vapour amount are fixed parameters in the model inversion, their cross interferences need to be quantified in order to exclude any significant influence on the methane measurement. The influence of local gradients (e.g. emission plumes or co-emission) of these gases, as well as aerosol induced light path variations (see above) should be quantified (e.g. by using the introduced model).*

The methane, $CO_2$, and water vapour are all retrieved parameters. We have revised the manuscript in Sections 4.4 and 4.5 in order to make this fact more clear.

3) *Figure. 3 (a) indicates that within the pass band of the order sorting filter there are three FP transmission fringes. This should be mentioned/motivated in the instrument description. Selecting 3 transmission peaks triples the light throughput compared to a measurement with a single peak and therefore enhances the SNR by sqrt(3). On the other hand, a dilution of the absorption signal of strong absorption lines is expected, reducing the SNR by up to a factor of 3. For the FP's free spectral range correlating with the spectral separation of strong periodic absorption structures (e.g. as in Vargas-Rodriguez and Rutt, 2009 or Kuhn et al., 2019) the sensitivity, selectivity and the SNR would be increased by using several FP transmission peaks. The instrument description does not mention if such a correlation is used.*

We include the motivation for the specific choice of spectral bandpass and number of F-P modes in Section 2.2..

The reviewer mentions interesting papers in which periodic structure between rovibrational transitions and FP transmission modes are exploited to enhance a combination of the signal and fractional absorption. GHGSat-D does not make use of this correlation.

4) *How does the measurement error/sensitivity vary across the imaging FOV? In Fig. C2 4 (a)-(d) the rings of equal FP transmission are faintly visible. A pixel located in the centre of the detector will see a different FP interferogram compared to a pixel close to the detector edge. I.e. the progression of the signal as shown in Fig. 4 (e) is dependent on the location of a pixel on the detector. A slight tilt of the FP in along track direction would increase the radii of the FP rings within the FOV. This could have the advantage of a better coverage of the whole FOV with similar FP interferograms and also it would increase the range of FP tuning (~cos(alpha)) per pixel. Thereby the spectral information of the measurement could be further enhanced and areas on the detector with low dynamics in spectral FP changes (e.g. the centre area of the detector) were avoided.*

The reviewer is right to point out that the measurement sensitivity varies across the imaging FOV. We have included additional discussion to Section 5 to note this fact.

5) *Fig. 3 (c) shows the 'instrument signal' as a function of the radius. Here it would be illustrative to show the differential instrument signal between a typical methane plume and a plume free region. Thereby the sensitivity of the method in terms of measured optical depth per methane amount would become more clear. Also the influence of typical CO2 and water vapour absorption gradients could be illustrated that way.*

The instrument signal in Figure 3(c) shows the response to the nominal background concentration levels of methane, CO2, and water vapour at a representative solar zenith angle and target elevation. We have revised the manuscript to include these exact values in the caption to Figure 3. Since the majority of the absorption features in instrument signal are due to methane, it is now possible to infer the sensitivity of the signal to an amount of methane.

**References:**

Houweling, S., Hartmann, W., Aben, I., Schrijver, H., Skidmore, J., Roelofs, G.-J. and Breon, F.-M.: Evidence of systematic errors in SCIAMACHY-observed CO 2 due to aerosols, Atmos. Chem. Phys., 5(11), 3003–3013, 2005.

Huang, Y., Natraj, V., Zeng, Z. and Yung, Y. L.: Quantifying the impact of aerosol scattering on the retrieval of methane from airborne remote sensing measurements, Atmos. Meas. Tech. Discuss., 1–28, 2020.

Thorpe, A. K., Frankenberg, C. and Roberts, D. A.: Retrieval techniques for airborne imaging of methane concentrations using high spatial and moderate spectral resolution: application to AVIRIS, Atmos. Meas. Tech., 7(2), 491–506, 2014.

---

## Author Comment (AC3) · 21 Dec 2020

**Responses to reviewers: "The GHGSat-D imaging spectrometer"**

We thank the reviewers for their comments and suggestions, which we address below. Reviewer comments are in *italics* and our responses are in normal font style.

**Anonymous Referee #2**

1) *P2, L25: "...which limits their detection sensitivity." Can you state any more quantitatively what the detection limit of these coarse spectral resolution instruments might be? Even providing a range would be helpful to the non-expert (like me).*

   Since the detection sensitivity not only depends on the spectral resolution, but on the specific spectral band and quantities such as the per-pixel signal level and absence of unwanted signal as well, it is hard to state a general quantitative relationship. Therefore, we have deleted this part of the sentence.

2) *P7, L21: It's becoming "standard" in our community to use the "Geoff Toon" solar transmittance spectrum. Mainly because everything thinks Geoff is incredibly careful and pays more attention to detail than anyone else. In any case, typically the best residuals are yielded when you use this linelist. It may not matter much for your application, but just FYI. (https://mark4sun.jpl.nasa.gov/toon/solar/solar_spectrum.html, I recommend 2016, diskintegrated spectrum). You have to multiply by the solar continuum yourself.*

   We switched to Geoff Toon's solar transmittance in the past few months ago and have noticed an improvement in retrieval quality. We thank the reviewer for the suggestion.

3) *P8, L25: I do not understand why it is so computationally expensive. Your forward model is literally Fsun(lambda) * mu0 * Albedo(lambda) /pi * exp(-m * tau(lambda)), where m is the airmass factor. Nearly all the work is done in calculating tau(lambda): tau(lambda) = tau_ch4(lambda) + tau_h2o(lambda) tau_ch4(lambda) = Sum_i=1..Nlayers{ sigma_ch4(i,lambda) * uch4(i) } tau_h2o(lambda) = Sum_i=1..Nlayers{ sigma_h2o(i,lambda) * uh2o(i) } The sigma's can be calculated once for the whole scene (unless you're retrieving surface pressure or temperature, which I do not think you are).*

   The computational expense results from the evaluation of the instrument + atmospheric model rather than just the atmospheric model itself. In particular, our instrument model contains an integral which is slow to compute.

4) *P9, L5: Is retrieving d analogous to retrieving the wavelength shift and stretch for a standard GOSAT-type retrieval? If so, say so.*

   We are not as familiar with GOSAT retrievals as the reviewer, but replacing a retrieval of $d$ with the wavelength equivalent would require not only a shift and stretch, but also a nonlinear component to account for the cosine dependence of the FP transmission mode position on the camera. The benefit of retrieving the FP gap spacing $d$ is that we are able to monitor this physical quantity over time to determine any mechanical or thermal drift.

5) *P9, Sect 4.5: It is hard to tell what you are doing without more information – could you please include a table that lists your state vector parameters, the source of your prior information or the prior values themselves, and the prior uncertainty you assume?*

As per Referee #1's comment (2), we have revised the manuscript to include information about what the atmospheric state parameters are. We have also revised the manuscript in Section 4.5 to mention that we currently set the prior variances large enough that the state parameter estimate is entirely determined by the data.

*Also, do you assume any spatial pixel-to-pixel correlations in your prior covariance matrix?*

We do not assume any spatial pixel-to-pixel correlations in the prior covariance matrix.

*Do you retrieve a scale factor for CH4, CH4 in the boundary layer only, or a full profile in some way? Etc. {It looks like you retrieve "VCD". But do you spread this equally in the vertical? Ie, you might have sensitivity to HOW you retrieve CH4. What if you put it all in the boundary layer? Because the gas absorption cross section is different down there, I expect it will affect your results in a small way.}*

We have revised the manuscript to include the fact that the molecular components of the Jacobian are calculated assuming an enhancement in the lowest atmospheric layer only.

*Also, I'm surprised you can use AIRS. Plus AIRS will die someday. And all the decent NWP models assimilate it anyway. Why not just use GEOS5 or ECMWF model output?*

We thank the reviewer for this suggestion. We will investigate incorporating these data sources.

*Additionally, where does your prior surface pressure and temperature profile come from?*

We state in Section 4.2 that we use the US_Standard pressure and temperature profile.

*Do you assume a fixed surface elevation across the field of view, or do you use a high-res DEM to get the surface elevation for each pixel, and a hypsometric adjustment to determine the prior surface pressure of each pixel? You don't say anything about if you can do this over complex topography or not.*

Yes, we assume a fixed surface elevation across the field of view. Given the GHGSat-D error levels, we are only sensitive to elevation changes of approximately >1km.

*Finally, can you say how many iterations this takes typically, and how long the retrieval takes for one full scene of data (on however many CPUs you typically use).*

We have revised the manuscript and included this information in Section 4.3.

*Same paragraph – do you have any way to test for and remove bad pixels? You could also attempt to include a stray light parameter or two in your forward model. You'd probably have the information content to solve for it.*

We do test and remove bad pixels, as noted in Section 4.6.  We thank the reviewer for the suggestion to try and retrieve stray light in our retrieval.

6) *P12, L10: I'm really surprised by this 6.5% theoretical (noise-drive) error. This is something like 120 ppb of methane. TROPOMI has errors more on the order of 10 ppb. And your number assumes PERFECT knowledge of temperature, water vapor, surface albedo, surface pressure, spectroscopy, instrument, etc. Literally everything but the instrument noise, which isn't bad; you list radiance SNR = 200 at these conditions. I have a feeling that if I wrote my own simple Beer's law forward model in wavelength space, and put an SNR of 200 per channel, using your spectral resolving power, I'd get a posterior uncertainty of less than 1%. What am I missing? In other words, can you say why this noise-driven uncertainty is so large compared to what a grating instrument with a similar SNR and spectral resolution would achieve?*

The difference between a grating instrument and our FP instrument is that we have multiple FP modes contributing to the signal at a given camera pixel, which can reduce the fractional absorption and hence methane sensitivity if not properly optimized, as is the case in GHGSat-D. In the recently launched GHGSat-C1, the FP spectroscopic configuration has been chosen such that shot-noise limited posterior error is indeed less than 1%.

---

## Author Response (AR2)

**Responses to reviewers: "The GHGSat-D imaging spectrometer"**

We thank the reviewers for their comments and suggestions, which we address below. Reviewer comments are in *italics* and our responses are in normal font style.

**Editor's Comments:**

1) *There is a special issue in AMT on the COMET campaign out of which some papers might fit into your introductory overview: https://amt.copernicus.org/articles/special_issue1034.html*

Thank you for alerting us to this special issue. We have added a reference to the airborne DIAL paper (Wolff et al., 2020) in Section 1.2, a methane measurement technique that was not previously cited in our introductory section.

10

2) *Reviewer 2 asks for a table of the state vector elements and their priors. Please also add such a table. Please add in section 4.2 the external parameter sources for pressure, temperature and topography (and also add related assumptions such as flat topography). While these assumptions might be uncritical at the current performance level of Claire. The new GHG-SAT generation migth require refinements in the forward model.*

15

 We have added Table 3 which lists the atmospheric state vector elements in our scene wide average retrieval, as well as their prior mean and standard deviations. We have added the source for the pressure, temperature, and target elevation in Section 4.2 as well as the assumption of flat topography within the field of view in Section 4.4.

20

3) *Reviewer 1 (her/his comment 5) asks for illustrating the effect of a typical plume enhancement in Figure 3 c. In my opinion, this would be a clear improvement. Please consider adding such an illustration (or give good reasons why this is not a good idea).*

We have added a methane Jacobian to Figure 3 to accommodate this request.

25

4) *Page 12, line 16: The posterior error formula contains the noise and the smoothing error contribution. It appears unusual to include the smoothing error for a noise assessment (or to use the full formula and to make the second term vanish artificially). Consider just using the definition of the propagated noise error "G Sy G^T", where G is the gain/contribution matrix.*

30 The reason we use the full formula is because the performance estimate requires that different state vector elements require different prior variances. We feel like it is instructive to start from the full formula and add motivation for why the prior variances are set the way they are (i.e. we assume perfect knowledge of all state vector elements save methane, and zero prior knowledge of methane itself).

**Anonymous Referee #1**

1) *Scattering at aerosol is neglected in the atmospheric model motivated by e.g. the work of Houweling et al., 2005, which treats total column CO2 measurements. I think that the finding of a decreased aerosol induced error for narrow layers close to Earth's surface cannot directly be transferred to the presented study of localised and strong emission plumes with much higher spatial resolution. The presented methane emission measurements show a different geometry with very high concentrations at low altitude. For low altitude aerosol and particularly for co-emitted aerosol (as mentioned in the manuscript, l.4,p.8), induced light path changes are likely to have a stronger impact on the vertical column density quantification. And thus, the influence of aerosol on the flux determination can become relevant.*

We agree with the reviewer that the case studied by (Houweling et al., 2005) is similar, but not exactly the same, to the case of a localized aerosol plume co-emitted with methane. Therefore, we have included a citation to a recent study (Huang et al., 2020) in Section 4.2 showing that neglecting aerosols leads to approximately 5% error for cases of a co-emitted aerosol plume with significant aerosol optical depth, an error level that is small compared to other errors that afflict our measurement (8-25%, as reported in Section 5). We have also included a citation of a AVIRIS methane retrieval paper (Thorpe et al., 2014) which justifies neglecting aerosol scattering in the forward model on similar grounds.

2) *The authors mention the absorption of CO2 and water vapour in the chosen wavelength window (p4, l13-14). If CO2 and water vapour amount are fixed parameters in the model inversion, their cross interferences need to be quantified in order to exclude any significant influence on the methane measurement. The influence of local gradients (e.g. emission plumes or co-emission) of these gases, as well as aerosol induced light path variations (see above) should be quantified (e.g. by using the introduced model).*

The methane, $CO_2$, and water vapour are all retrieved parameters. We have revised the manuscript in Sections 4.4 and 4.5 in order to make this fact more clear.

3) *Figure. 3 (a) indicates that within the pass band of the order sorting filter there are three FP transmission fringes. This should be mentioned/motivated in the instrument description. Selecting 3 transmission peaks triples the light throughput compared to a measurement with a single peak and therefore enhances the SNR by sqrt(3). On the other hand, a dilution of the absorption signal of strong absorption lines is expected, reducing the SNR by up to a factor of 3. For the FP's free spectral range correlating with the spectral separation of strong periodic absorption structures (e.g. as in Vargas-Rodriguez and Rutt, 2009 or Kuhn et al., 2019) the sensitivity, selectivity and the SNR would be increased by using several FP transmission peaks. The instrument description does not mention if such a correlation is used.*

We include the motivation for the specific choice of spectral bandpass and number of F-P modes in Section 2.2..

The reviewer mentions interesting papers in which periodic structure between rovibrational transitions and FP transmission modes are exploited to enhance a combination of the signal and fractional absorption. GHGSat-D does not make use of this correlation.

4) *How does the measurement error/sensitivity vary across the imaging FOV? In Fig. C2 4 (a)-(d) the rings of equal FP transmission are faintly visible. A pixel located in the centre of the detector will see a different FP interferogram compared to a pixel close to the detector edge. I.e. the progression of the signal as shown in Fig. 4 (e) is dependent on the location of a pixel on the detector. A slight tilt of the FP in along track direction would increase the radii of the FP rings within the FOV. This could have the advantage of a better coverage of the whole FOV with similar FP interferograms and also it would increase the range of FP tuning (~cos(alpha)) per pixel. Thereby the spectral information of the measurement could be further enhanced and areas on the detector with low dynamics in spectral FP changes (e.g. the centre area of the detector) were avoided.*

The reviewer is right to point out that the measurement sensitivity varies across the imaging FOV. We have included additional discussion to Section 5 to note this fact.

5) *Fig. 3 (c) shows the 'instrument signal' as a function of the radius. Here it would be illustrative to show the differential instrument signal between a typical methane plume and a plume free region. Thereby the sensitivity of the method in terms of measured optical depth per methane amount would become more clear. Also the influence of typical $CO_2$ and water vapour absorption gradients could be illustrated that way.*

The instrument signal in Figure 3(c) shows the response to the nominal background concentration levels of methane, $CO_2$, and water vapour at a representative solar zenith angle and target elevation. We have revised the manuscript to include these exact values in the caption to Figure 3. Since the majority of the absorption features in instrument signal are due to methane, it is now possible to infer the sensitivity of the signal to a given amount of methane.

**Anonymous Referee #2**

1) *P2, L25: "…which limits their detection sensitivity." Can you state any more quantitatively what the detection limit of these coarse spectral resolution instruments might be? Even providing a range would be helpful to the non-expert (like me).*

Since the detection sensitivity not only depends on the spectral resolution, but on the specific spectral band and quantities such as the per-pixel signal level and absence of unwanted signal as well, it is hard to state a general quantitative relationship. Therefore, we have deleted this part of the sentence.

2) *P7, L21: It's becoming "standard" in our community to use the "Geoff Toon" solar transmittance spectrum. Mainly because everything thinks Geoff is incredibly careful and pays more attention to detail than anyone else. In any case, typically the best residuals are yielded when you use this linelist. It may not matter much for your application, but just FYI. (https://mark4sun.jpl.nasa.gov/toon/solar/solar_spectrum.html, I recommend 2016, diskintegrated spectrum). You have to multiply by the solar continuum yourself.*

We switched to Geoff Toon's solar transmittance in the past few months ago and have noticed an improvement in retrieval quality. We thank the reviewer for the suggestion.

3) *P8, L25: I do not understand why it is so computationally expensive. Your forward model is literally Fsun(lambda) \* mu0 \* Albedo(lambda) /pi \* exp(-m \* tau(lambda)), where m is the airmass factor. Nearly all the work is done in calculating tau(lambda): tau(lambda) = tau_ch4(lambda) + tau_h2o(lambda) tau_ch4(lambda) = Sum_i=1..Nlayers{ sigma_ch4(i,lambda) \* uch4(i) } tau_h2o(lambda) = Sum_i=1..Nlayers{ sigma_h2o(i,lambda) \* uh2o(i) } The sigma's can be calculated once for the whole scene (unless you're retrieving surface pressure or temperature, which I do not think you are).*

The computational expense results from the evaluation of the instrument + atmospheric model rather than just the atmospheric model itself. In particular, our instrument model contains an integral which is slow to compute.

4) *P9, L5: Is retrieving d analogous to retrieving the wavelength shift and stretch for a standard GOSAT-type retrieval? If so, say so.*

We are not as familiar with GOSAT retrievals as the reviewer, but replacing a retrieval of $d$ with the wavelength equivalent would require not only a shift and stretch, but also a nonlinear component to account for the cosine dependence of the FP transmission mode position on the camera. An additional benefit of retrieving the FP gap spacing $d$ is that we are able to monitor this physical quantity over time to determine any mechanical or thermal drift.

5) *P9, Sect 4.5: It is hard to tell what you are doing without more information – could you please include a table that lists your state vector parameters, the source of your prior information or the prior values themselves, and the prior uncertainty you assume?*

As per Referee #1's comment (2), we have revised the manuscript to include information about what the atmospheric state parameters are. We have also revised the manuscript in Section 4.5 to mention that we currently set the prior variances large enough that the state parameter estimate is entirely determined by the data.

*Also, do you assume any spatial pixel-to-pixel correlations in your prior covariance matrix?*

We do not assume any spatial pixel-to-pixel correlations in the prior covariance matrix.

*Do you retrieve a scale factor for CH4, CH4 in the boundary layer only, or a full profile in some way? Etc. {It looks like you retrieve "VCD". But do you spread this equally in the vertical? Ie, you might have sensitivity to HOW you retrieve CH4. What if you put it all in the boundary layer? Because the gas absorption cross section is different down there, I expect it will affect your results in a small way.}*

We have revised the manuscript to include the fact that the molecular components of the Jacobian are calculated assuming an enhancement in the lowest atmospheric layer only.

*Also, I'm surprised you can use AIRS. Plus AIRS will die someday. And all the decent NWP models assimilate it anyway. Why not just use GEOS5 or ECMWF model output?*

We thank the reviewer for this suggestion. We will investigate incorporating these data sources.

*Additionally, where does your prior surface pressure and temperature profile come from?*

We have revised he manuscript to state in Section 4.2 that we use the US_Standard pressure and temperature profile.

*Do you assume a fixed surface elevation across the field of view, or do you use a high-res DEM to get the surface elevation for each pixel, and a hypsometric adjustment to determine the prior surface pressure of each pixel? You don't say anything about if you can do this over complex topography or not.*

Yes, we assume a fixed surface elevation across the field of view. Given the magnitude of GHGSat-D error levels, we are only sensitive to elevation changes of approximately >1km.

*Finally, can you say how many iterations this takes typically, and how long the retrieval takes for one full scene of data (on however many CPUs you typically use).*

We have revised the manuscript and included this information in Section 4.3.

*Same paragraph – do you have any way to test for and remove bad pixels? You could also attempt to include a stray light parameter or two in your forward model. You'd probably have the information content to solve for it.*

We do test and remove bad pixels, as noted in Section 4.6. We thank the reviewer for the suggestion to try and retrieve stray light in our retrieval.

6) *P12, L10: I'm really surprised by this 6.5% theoretical (noise-drive) error. This is something like 120 ppb of methane. TROPOMI has errors more on the order of 10 ppb. And your number assumes PERFECT knowledge of temperature, water vapor, surface albedo, surface pressure, spectroscopy, instrument, etc. Literally everything but the instrument noise, which isn't bad; you list radiance SNR = 200 at these conditions. I have a feeling that if I wrote my own simple Beer's law forward model in wavelength space, and put an SNR of 200 per channel, using your spectral resolving power, I'd get a posterior uncertainty of less than 1%. What am I missing? In other words, can you say why this noise-driven uncertainty is so large compared to what a grating instrument with a similar SNR and spectral resolution would achieve?*

The difference between a grating instrument and our FP instrument is that we have multiple FP modes contributing to the signal at a given camera pixel, which can reduce the fractional absorption and hence methane sensitivity if not properly optimized, as is the case in GHGSat-D. In the recently launched GHGSat-C1, the FP spectroscopic configuration has been chosen such that shot-noise limited posterior error is indeed less than 1%.